# Headedness and the Lexicon: The Case of Verb-to-Noun Ratios

**Maria Polinsky [1],\***  **and Lilla Magyar [2]**

[1]  Department of Linguistics, University of Maryland, College Park, MD 20742-7505, USA
[2]  Independent Scholar, Cambridge CB19YP, UK; lillamagyar0929@gmail.com
\*  Correspondence: Polinsky@umd.edu

**Abstract:** This paper takes a well-known observation as its starting point, that is, languages vary with respect to headedness, with the standard head-initial and head-final types well attested. Is there a connection between headedness and the size of a lexical class? Although this question seems quite straightforward, there are formidable methodological and theoretical challenges in addressing it. Building on initial results by several researchers, we refine our methodology and consider the proportion of nouns to simplex verbs (as opposed to light verb constructions) in a varied sample of 33 languages to evaluate the connection between headedness and the size of a lexical class. We demonstrate a robust correlation between this proportion and headedness. While the proportion of nouns in a lexicon is relatively stable, head-final/object-verb (OV)-type languages (e.g., Japanese or Hungarian) have a relatively small number of simplex verbs, whereas head-initial/verb-initial languages (e.g., Irish or Zapotec) have a considerably larger percentage of such verbs. The difference between the head-final and head-initial type is statistically significant. We, then, consider a subset of languages characterized as subject-verb-object (SVO) and show that this group is not uniform. Those SVO languages that have strong head-initial characteristics (as shown by the order of constituents in a set of phrases and word order alternations) are characterized by a relatively large proportion of lexical verbs. SVO languages that have strong head-final traits (e.g., Mandarin Chinese) pattern with head-final languages, and a small subset of SVO languages are genuinely in the middle (e.g., English, Russian). We offer a tentative explanation for this headedness asymmetry, couched in terms of informativity and parsing principles, and discuss additional evidence in support of our account. All told, the fewer simplex verbs in head-final/OV-type languages is an adaptation in response to their particular pattern of headedness. The object-verb/verb-object (OV/VO) difference with respect to noun/verb ratios also reveals itself in SVO languages; some languages, Chinese and Latin among them, show a strongly OV ratio, whereas others, such as Romance or Bantu, are VO-like in their noun/verb ratios. The proportion of nouns to verbs thus emerges as a new linguistic characteristic that is correlated with headedness.

**Keywords:** lexical categories; nouns; verbs; syntax; structural head

## 1. The Starting Point

People learning Japanese as a second language have to struggle with many aspects of its grammar and orthography but there is at least one area of Japanese that is relatively easy to an L2 learner, i.e., the formation of verbs. L2 learners of Japanese quickly come to realize that not knowing a verb does not prevent them from being understood. When at a loss, an L2 learner takes a noun, combines it with the verb *suru* "do" and the result is comprehensible, even though it may not quite be idiomatic Japanese. These days, *suru* often combines with English words, such as *beesu appu suru* "increase salary" (from *base up*), *kisu suru* "kiss" (from kiss) *guuguru suru* "Google", and many others. Although this trend is

accompanied by the usual hand-wringing about the Japanese vocabulary being destroyed by English, *suru* has, in fact, been used with Mandarin Chinese loans in a similar way for centuries, yielding such compounds as *kenkyuu suru* "study" and *gensyoo suru* "decrease".[1] This initial observation suggests that Japanese has a rather small number of inflecting verbs and a large open class of complex predicates. Such complex predicates are created from non-verbal constituents combined with light verbs, of which *suru* is the most common.

Turning to lesser-studied languages, Pawley (2006) discusses the northern Australian language Djamindjung (djd) and the Papua New Guinean (PNG) language Kalam (kmh), which both have a real paucity of verb roots, just over a hundred. These small, closed classes of verb roots occur as independent verbs, but all other verb meanings are expressed by complex predicates, as in Japanese. Pawley suggests that these languages are not unique, and that related Australian and PNG languages also have small, closed verb classes.

What do other languages do? The response of English to the need for new verbs is to directly convert words of other classes into verbs without any additional morphology, yielding such expressions as *to ftp*, to R the data, to KCCO a friend, or, from the days of the Clinton White House, *to Linda-Tripp someone*. More morphologically prolific languages build new inflected verbs using affixation, for example, modern Russian has experienced a considerable influx of English words, in the last few decades, and has been creating verbs, in droves, such as *piarit* "to PR", *parkovat* "park", *postí* "post on a blo"', or *kopipejstit* "copy and paste".

Therefore, the difference between English and Russian, on the one hand, and Japanese, Djamindjung, and Kalam, on the other hand, is that, while the former languages freely create new verb roots or stems to add new verbal entities to the lexicon, the latter three do not; instead, they rely on light verbs to produce new complex verbs. The three languages that utilize light verbs are head-final or SOV type languages. Is this an accident, or is this scarcity of inflecting verbs somehow related to headedness? This is the essence of the main question that we explore in this paper:

(1)    Does the verb-to-noun ratio differ across headedness types?

If the informal observations above are not illusory, we would expect to find a lower proportion of verbs to nouns in head-final languages. To anticipate the discussion below, we build on and expand the initial generalizations first proposed by Polinsky (2012) and, then, further explored by Seifart et al. (2014) and Strunk et al. (2015).

The rest of the paper is structured as follows: Section 2 explains the basic notions that this paper relies on parts of speech (lexical categories) and headedness, with particular subtypes of headedness considered in this study. In Section 3 we explain the details of our data collection and discuss some of the practical problems that arise in that regard. Not all of them can be resolved in a perfect way, but we offer some considerations in support of the existing solutions. Section 4 presents our main hypothesis concerning the distribution of nouns and verbs with respect to headedness and offers quantitative support for this hypothesis based on head-final and head-initial languages. In Section 5 we turn to languages with the subject-verb-object (SVO) order in the main clause and based on this order alone, one would expect them to be head-initial types. However, we show that the SVO type is not uniform; instead, SVO languages fall into separate headedness subgroups, and their distribution of nouns and verbs correlates well with other properties associated with head-final or head-initial structures. Finally, Section 6 offers a tentative explanation of the patterns that we document in this work.

---

[1]    The Mandarin Chinese borrowings are so numerous that it would take a number of pages to list just part of them; see (Jacobsen 1992, chp. 7) for representative examples.

## 2. Basic Notions

In this section, we define nouns and verbs, the main lexical classes (parts of speech) that we are concerned with and discuss headedness classes. As with any operational definitions, it is impossible to find definitions that would satisfy everyone, but in working toward that goal, we have tried to stay as theory neutral and inclusive as possible.

### 2.1. Verbs and Nouns

A verb is understood, here, as a member of the syntactic class of words that is inherently predicative, constitutes a minimal predicate in a clause, takes arguments (subject in particular), and can be inflected for tense, aspect, mood, and agreement (Langacker 1987; Baker 2003).

Most linguists have historically agreed that all languages have some universal structural building blocks, among which are the lexical categories of nouns and verbs. However, "a persistent thread of research that maintains that there are languages that do not have … familiar … categories" (Chung 2012) has created serious doubts about this universality. While the division of the lexicon into nouns and verbs is likely universal, the diagnostics for lexical classes are language specific and can be highly obscure or subtle. In general, the identification of nouns vs. verbs relies on formal patterns of inflection, morphological derivation, and syntactic distribution (Schachter 1985; Sasse 1993; Baker 2003; Kaufman 2009; Chung 2012).

The languages for which a stringent lexical division between nouns and verbs has been most doubted are characterized by a large class of roots that can be used either nominally or verbally. Representative examples include Tongan (Broschart 1997), Mandarin Chinese (Chao 1968), Riau Indonesian (Gil 2005), and Mapudungun (Malvestitti 2006). Such categorially ambiguous languages often have polysynthetic features (Lois and Vapnarsky 2006 for Amerindian, Aranovich 2013 for Fijian, Arkadiev et al. 2009 for Adyghe) or templatic morphology (Arad 2003) and include many multifunctional content words.

A careful analysis of categorially ambiguous content words usually shows fine-grained distinctions that permit the desired differentiation of lexical categories. For example, in Adyghe, only proper nouns can co-occur with possessive affixes; derived nouns (e.g., nouns derived from verbs) cannot (Arkadiev et al. 2009, p. 32), as seen here:[2]

(2)   Adyghe

a.   s-jə-pəj

 1sg-poss-enemy

 "my enemy"

b.   *s-jə-k$_w$a-ʁe

 1sg-poss-go-pst

 "my departed"

Although this is not the place to go into details, much research has shown that the distinctions between nouns and verbs are always available, although they are less obvious in some languages as compared with other languages. Chung (2012)'s meticulous study, for example, uses subtle but reliable diagnostics to argue that Chamorro has noun, verb, and adjective categories. Other studies that identify fine-grained distinctions between nominal and verbal roots include Arad (2003) semantic analysis of the relations between nominal and verbal roots in Hebrew (showing principled rules underlying root polyvalence) and Haviland (1994) analysis of roots in Tzotzil (Tsotsil).

---

[2]   The abbreviations follow the Leipzig Glossing Rules. Additional abbreviations: LVC—light verb construction, POS—part of speech.

Furthermore, recent experimental work attests to processing differences between verbs and other parts of speech (Vigliocco et al. 2011). These results are pleasing in that they provide novel support to the theoretical conception that predication is a fundamental property of language design and that verbs are the vehicle of this property.[3]

For the purposes of this paper, we have assumed that the lexicon of a given language is divided into nouns and verbs based on language-particular criteria, including inflectional morphology, semantic correspondences (Arad 2003; Chung 2012), and syntactic distribution. In some of the languages cited below, most notably Zinacantec Tzotzil (Haviland 1994); Hiaki (Harley 2014); and Nishnaabemwin (Thivierge 2018), the noun–verb division is established at the level of roots rather than lexical items.[4]

Assuming that verbs are a fundamental category of language design, there are different ways of forming verbs and verb phrases, from using words in an open class of lexical items (primary lexical verbs) to using a combination of nonverbal material (often a noun) and a verb with a rather general meaning ("become", "give", "do", "make") to form a light verb construction, LVC below (Grimshaw and Mester 1988; Osborne and Groß 2012). This is where languages differ significantly, whereas most languages allow LVCs, some languages rely on LVCs much more heavily than other languages, as illustrated by the Japanese examples in the preceding section.)

In the discussion below, we focus on the content of the verb class in a given language as represented by lexical verbs, to the exclusion of LVCs. That does not necessarily mean that all the verbs we consider are going to be simplex. A great deal depends on the morphology of a given language, the means that can be deployed to derive verbs, and the boundaries between derivation and inflection in the verbal domain. The distinction between roots, stems, and lexical items is also relevant, and it is impossible to apply it wholesale to all the languages. In order for our study to proceed, at least initially, we made a conscious decision to rely on researchers who worked on a particular language in order to determine what counts as a lexical verb and what counts as a lexical noun. To recapitulate, the criteria can differ across languages, but even with that difference, it is reasonable to expect some consistency in the results.

The definition of a noun seems less elusive than that of a verb. Nouns are defined distributionally as elements that can appear in argument positions and as complements of adpositions. Nouns can inflect for number, case, gender, possession, and definiteness/specificity. As with verbs, we rely on studies of a particular language for their identification of nouns in that language and for the exclusion of dubious elements.

## 2.2. Headedness and Headedness Types

Before we proceed, a clarification is on order. Throughout the paper, we consider headedness primarily in terms of the structure of the verb phrase (vP/VP), noun phrase (NP/DP), and adpositional phrase (PP). This makes headedness, as discussed here, a notion that is narrower than "basic word order", which is established at the level of a clause or utterance. Word order in an utterance is subject to such factors as information structure. By narrowing our conception of headedness, we can achieve more precise results. However, in linguistic literature, it is common to use such labels as "head-final" and "SOV" interchangeably, and we cannot abandon this practice easily. However, it is accepted that some orders seem to be more sensitive to the general headedness of a given language than others; the "sensitive" orders include the order of a relative clause and head noun, the position of the predicate in an embedded clause, and the order of adposition and its complement. The reader should bear

---

[3]  It is often stated that nouns are easier to process than verbs (Vigliocco et al. 2011; Szekely et al. 2005). However, in a recent paper, Seifart et al. (2018) show that in eight languages out of nine in their sample, pauses are longer before nouns than before verbs, which they interpret as a sign of additional processing difficulty associated with nouns. English, the ninth language in their sample, shows the opposite trend (more pauses before verbs). The eight other languages in their sample are pro-drop, which may be a confounding factor in the results.

[4]  We will return to the issue of diagnostics and numerical estimates in Section 3.

in mind that the order of constituents in a clause is subject to more factors than considered here; accordingly, the traditional labels "SOV", "VSO", "SVO" are used as a shorthand. With this clarification in place, we now turn to a consideration of word orders in terms of headedness.

As far as headedness goes, the main contrast is between head-final and head-initial languages. Within the head-final type, languages such as Japanese and Korean are "rigidly head-final" types (Kayne 1994; Siewierska 1997, a.o.), meaning that their heads consistently follow their dependents in all types of phrases (although even these languages allow some postverbal material in the root clause).

Languages such as German or Persian can be considered exemplars of the nonrigid head-final type; their head-final property seems to be a violable constraint. In reality, few languages are strictly head-final or head-initial types, the orders that can be characterized as "harmonic" or rigid (Hawkins 1983; Biberauer and Sheehan 2013; Sheehan et al. 2017; Dryer 2013a, 2013b).

Headedness principles also apply in an asymmetrical manner, and that is manifested in two ways, long noted in cross-linguistic investigations, starting with Greenberg's work. First, rigidly head-final languages do not allow verb-medial or verb-initial orders, whereas head-initial languages (VSO, VOS) do not exhibit a similar (inverse) rigidity. In other words, verb-initial languages always seem to allow verb-medial (SVO) orders and those that do not do so are either impossible or rare (Greenberg 1963; Siewierska 1997). Second, natural languages do not allow a head-initial phrase that is contained in a head-final phrase in the same structural domain, whereas the opposite containment is possible. This constraint has become known as Final-over-Final Condition (FOFC), and Sheehan et al. (2017) argue that FOFC is a linguistic universal, not just a tendency or a constraint on processing.

Once we consider all these additional complications, the classification of a given language in terms of headedness can become confusing. For instance, is Yucatec Mayan VOS or SVO? Its most frequent word order is SVO; however, all its genetic relatives are verb-initial, and it still uses a number of verb-initial orders. Understandably, researchers cannot agree: Briceño (2002) and Gutiérrez Bravo and Monforte (2008, 2009) classify Yucatec Mayan as SVO; Hofling (1984) and Durbin and Ojeda (1978) argue that it has two basic word orders, SVO and VOS, but with a secondary statistical preference for SVO; finally, Gutiérrez-Bravo and Monforte (2010) suggest that it is SVO with two-place predicates and VS in objectless clauses. This type of inconsistency in headedness is attested cross-linguistically (Dryer 2013a). It may be possible to overcome doubts regarding a particular language in a focused study of a given language, but in a larger cross-linguistic survey of the type intended here, coarser characterizations are inevitable. Either a number of languages have to be classified as having a "mixed" word order (Seifart et al. 2014; Strunk et al. 2015), or one of their word orders has to be selected to the exclusion of others.

In establishing the subtypes for our query, we have tried to balance the need to recognize different headedness subtypes with the desire to have as few types as possible. The types presented here were checked against the typology proposed by Matthew Dryer. The identification of headedness was based on the order of subject, object, and verb; order of relative clause and head noun (Dryer 2013b); the relationship between the order of object and verb and the order of adposition and its complement (Dryer 2013c), and the relationship between the order of object and verb and the order of relative clause and head noun (Dryer 2013d). On the basis of these criteria, languages can be divided into head-final (rigid and non-rigid) and head-initial types. For the latter type, we choose languages that not only allow head-initial orders in phrases other than a clause but also have verb-initial orders in the matrix clause (VSO, VOS). Notably, while SOV languages can be rather rigid with respect to the order of constituents in a clause, VSO and VOS languages always allow SVO as an alternative (Dryer 2013a).

These types and languages that instantiate them are shown in Table 1.

Within this broad-based typology, as the first pass, we would like to examine the ratio of verbs to nouns in languages illustrating each type. To recapitulate, our main question is stated in Section 1, (1) above, that is, Does the ratio of nouns and simplex verbs differ across the headedness types?

In order to answer this question, we collected quantitative data on the number of nouns and verbs in a language sample that included head-initial and head-final languages and compared these two types.

Assuming that the percentage of nouns in a lexicon is relatively stable (something that we show to be correct in the data below), the predictions of this comparison are as follows:

**Table 1.** Basic headedness types used in this study.

|  | Head-Final | | Head-Initial |
|---|---|---|---|
|  | **Rigid** | **Non-Rigid** | |
| **Example languages** | Japanese, Korean | Basque, Latin, Persian | Irish, Malagasy, Tzotzil |

(3)   Predictions regarding lexicon and headedness

**Null hypothesis:** Head-final and head-initial languages have a comparable distribution of nouns and verbs;

**Alternative hypothesis:** Head-final languages have a lower ratio of verb types to noun types.

The next three sections present the results of our quantitative investigation into the ratio of nouns and verbs vis-à-vis headedness. Before we delve further into the comparison of language types, we discuss how the basic data for this comparison is collected.

## 3. Data Collection

### 3.1. General Remarks

In order to get a set of comparable data, we have limited this part of our query to the ratio of verbs to nouns. This seems a reasonable measure because if adverbs and adjectives are also included, the two other lexical categories that are often included in the counts, we would start losing the strength of the cross-linguistic comparison. While noun–verb distinctions may sometimes be subtler than is traditionally assumed (as discussed in Section 2.1), all languages have nouns and verbs. However, not all languages have easily identifiable adjectives and adverbs, which is another reason to exclude them (see Dixon 1982, Baker 2003).

The first and foremost need we addressed in building our language sample had to do with creating a balance across the relevant types identified in Table 1 above. Assuming that, as our main goal, we also tried to maintain genetic and areal balance, to the degree that was possible given other limitations that we will now turn to.

Our investigation is naturally confined by the available data on texts and dictionaries representing different languages of interest to us. Although English and some other Indo-European languages are catalogued in WordNet (Miller et al. 1990) or CELEX, data on other languages is much more limited and surprisingly hard to come by. Furthermore, WordNet and CELEX are not sufficiently comparable across languages and are structured in ways that prevent good cross-linguistic comparison. Some of the problems in comparing WordNet and CELEX data are discussed by Polinsky (2012) and are further elaborated on by Strunk et al. (2015). The sample collected and analyzed, here, was designed to circumvent these problems.

### 3.2. Dictionary or Corpus?

Wherever possible, we drew data from the materials on universal dependencies (UD) collected from treebanks in multiple languages (http://universaldependencies.org). This platform allows us to collect data on the distribution of lexical categories (parts of speech, POS) across different languages. The project itself is admirable and as it grows, it will offer more data. The most attractive component

of this project has to do with the uniformity of morphological analysis, especially as applied to core lexical categories. However, this project can only work with what is available. Some corpora are still tagged better than others, for example, some treebanks are still missing morphological analyses (for example, Telugu is in the UD bank, but it does not have a morphological analyzer); for some languages, POS counts vary depending on a particular subcorpus (see footnote 17), and finally, the size of a particular corpus matters (see footnote 9). Another helpful collection of corpora, some of them tagged for POS, is available at the Leeds corpus site assembled by Serge Sharoff (http://corpus.leeds.ac.uk/list.html); here too, the tagging is not always uniform, and establishing POS frequencies requires additional compilation and analysis.

Some languages in our sample have larger corpora, which would allow one to distinguish spoken and written language. However, for some other languages no such division is available, therefore, we purposely included all contemporary genres in the lemma search (historical texts were excluded, however). This measure is also motivated by the need to compare lemmas culled from corpora to data from dictionaries. The latter do not always divide lexica into spoken-language and written-language words, at least not consistently.

A study such as ours would have to base itself on comparable data for each language, ideally, corpora with a comparable composition in terms of genres, as they represent connected speech from different walks of life (see also Seifart et al. 2014; Strunk et al. 2015). We agree that this is an ideal goal, but it is hard to achieve with the present set of data tools. However, if we compare the verb-to-noun ratio across different genres in English, for which the data are quite rich, and which enjoys the luxury of good morphological analyzers, the result is more or less consistent.[5] As shown in Table 2, with the percentages from four different English subcorpora, drawn from Gold Standard Universal Dependencies Corpus (https://catalog.ldc.upenn.edu/LDC2012T13), the difference in the verb-to-noun ratio across genres is not significant ($p > 0.1$).

**Table 2.** Distribution of nouns and verbs across genres in English.[8]

| Verbs (%) | Nouns (%) | Verb-to-Noun Ratio | Corpus Size, Tokens | Genres |
|---|---|---|---|---|
| 11 | 33 | 0.33 | 254,829 | news, wiki |
| 15 | 36 | 0.41 | 80,041 | blog, social, reviews, email |
| 16 | 38 | 0.41 | 49,616 | academic, fiction, nonfiction, news, spoken, web, wiki |
| 15 | 36 | 0.41 | 21,540 | news, wiki |

Corpora are not available for all languages, therefore, we also had to consult dictionaries. That allowed us to include typologically representative data from more relevant languages. A direct comparison between lemma-based POS data from a corpus and data from a dictionary seems undesirable, and in an ideal world, it should probably be avoided.[6] To make sure that the use of corpus and dictionary data is legitimate (even if less than ideal), we compared data on the ratio of nouns and verbs in a Russian online corpus with the ratio based on the number of nouns and verbs in the grammatical dictionary by Zaliznjak (1977 and subsequent editions). In the UD Russian corpus, the number of noun lemmas is 16,761 and the number of verbs is 7832, so the verb-to-noun ratio is 0.47. Zaliznjak's dictionary, which includes about 106,000 words, has 55,694 nouns and 27,863 verbs. The verb-to-noun ratio is 0.50, almost identical to the ratio in the corpus.

In a similar evaluation, we compared the ratio of nouns and verbs in the UD corpus of English with the ratio based on the Oxford Dictionary (171,476 words, including obsolete words), which has

---

[5]   Previous studies (Polinsky 2012; Seifart et al. 2014; Strunk et al. 2015) presented the noun-to-verb ratio. Here we reverse it, primarily for expository purposes—as a reflection of the relative stability of the class of nouns across languages (see Section 4.2).

[8]   To reiterate, our study is based on types, however, corpus size is normally indicated in tokens, as shown in this table.

[6]   Dictionaries are probably more reliable for a study like ours, because the parts of speech are annotated by hand and do not depend on the vagaries of morphological analyzers or genre distribution.

92,597 nouns and 31,388 verbs. The ratio based on the online corpus data is 0.33, the one based on the dictionary is also 0.33 (see also Table 2).[7] This confirms the proof of principle, that is, corpus data and dictionary data can be legitimately compared for the purposes of this study. We would like to underscore the targeted nature of such a comparison, we are certainly not claiming that corpus data and dictionary data are comparable on all fronts and dimensions.

In working with dictionary data, we counted dictionary entries, which is equivalent to lemmas, unless the dictionary was specifically organized by roots, as with some languages mentioned above. In our sample, that is the case with Malagasy (Diksionera 1973; Abinal and Malzac 1899) and Zinacantec Tzotzil (Haviland 1994). Our verb counts included auxiliaries; some languages have those, and some do not, and for languages of the former type, all verbs, including auxiliaries, were included in the counts. However, the counts exclude compound verbs formed using a light verb, as in the Japanese LVCs above.

Speaking of comparability across languages, one expects great variance in the number of verb types in Slavic languages depending on whether perfective and imperfective verbs are assigned to the same verb type or two (similar issues can arise with derivational morphology in other languages). The Slavic verb issue is perennially difficult, not just with respect to this study, but with respect to general work on verb derivation and the structure of Slavic aspect (Borik 2006; Filip 2012; Gvozdanović 2012; Schoorlemmer 1995, for insightful discussion). In our counts for Slavic, we had to rely on the principles of identification abided by in a particular dictionary or corpus. (We have more to say on this issue in Section 5).

All in all, we had to rely on language-specific criteria to identify nouns and verbs, but we made a conscious effort to minimize the effect of language specificity, and therefore uniformly relied on type (not token) counts as robust frequency effects are attested when one looks at type frequencies (Bybee 1995).

Likewise, although data for individual languages vary both in quality and quantity, we tried to minimize the differences by using appropriate statistics (see Section 4.2) and balanced our sample to achieve a comparable number of languages in each type.

As this discussion shows, the data collection for the languages represented below was not always straightforward. Although more and more corpora for different languages are being collected and compiled, the actual materials included in the corpora do not always match, and morphological analyses and tagging principles vary widely. Further cooperation between theoreticians and lexicographers is of critical importance in this domain. Just as comparative syntax received a big boost from the micro-comparative work on closely related languages (Romance; Germanic; Semitic), so to, micro-comparative corpus building could lead to important breakthroughs that would benefit the field as a whole.

---

[7]  A crucial difference between corpora and dictionaries has to do with the percentage of parts of speech in the relevant collection. To continue with our side-by-side evaluation of corpora and dictionaries, compare the percentages of nouns and verbs in corpora and dictionaries of Russian and English (percentages rounded off to integers).

| | **Russian** | | **English** | |
|---|---|---|---|---|
| | **UD corpus** | **Dictionary** | **UD corpus** | **Dictionary** |
| **Verbs** | 17 | 26 | 17 | 25 |
| **Nouns** | 37 | 52 | 33 | 54 |

In this percentage count, the size of a corpus seems to play a significant role. For some languages in the UD corpus, the corpora are very small (for instance, the Irish corpus has 23,964 tokens, the Greek corpus, 61,773 tokens). With such small tagged corpora, it may be more expeditious to rely on dictionaries. In the representation of the quantitative data below, we will be showing the size of the corpus/dictionary as well.

## 4. Head-Initial and Head-Final Types

We repeat below our predictions concerning lexicon and headedness.

(4)    Predictions regarding lexicon and headedness

a.    **Null hypothesis**: head-final and head-initial languages have comparable distribution of nouns and verbs;

b.    **Alternative hypothesis**: Head-final languages have a smaller number of simplex verbs.

### 4.1. Head-Initial and Head-Final Types: Quantitative Data

To evaluate these predictions, we compared verb-to-noun ratios in the following languages that fit the characteristics of head-final or head-initial type more or less clearly. Our initial comparisons only target the distinction between the two types, without the recognition of "rigid" and "non-rigid" word order subtypes (Kayne 1994; Siewierska 1997, a.o.). This basic distinction is pursued for several reasons. First, we would like to test our predictions on the largest possible sample. Second, the recognition of a language as rigidly or non-rigidly X is based on the order of constituents in an utterance, and as we stated earlier, our main focus is on the order of elements in the verb phrase, noun phrase, and adpositional phrase. The order of constituents in an utterance is modulated, among other factors, by information-structural principles, which we do not consider here. These in turn, can color a particular researcher's assessment of a given language as having a rigid or nonrigid word order.

To reiterate, in determining the numerical values for the languages in the sample, we based the data on POS counts (in lemmas) from corpora or dictionaries and, then, measured the relative ratio of the number of verbs to the number of nouns. This reflects the general focus on types adopted in this paper. In that regard, our approach is different from the evaluation method used by Seifart et al. (2014) and Strunk et al. (2015), who focused on tokens and, accordingly, counted noun-to-verb ratio per each clause (N/(N+V)). This is an interesting approach, but we avoided it for a couple of reasons. First, as we already indicated, some languages are represented by dictionaries, which make token counts untenable. As we indicated in Section 3.2, data from corpora and data from dictionaries are in fact comparable with regard to verb types. Second, the corpora used by Strunk et al. (2015: Slide 3) are rather small, smaller than most of the corpora considered in this study. Finally, the clause-based approach makes it more difficult to compare languages with pro-drop (where we could expect a lower number of nouns) and the ones without pro-drop.

Table 3 presents data on the languages considered in our study, arranged with respect to headedness.

**Table 3.** Verbs and nouns across languages of the head-final (HF) and head-initial (HI) type.

| Language | Headedness | Verbs, Raw Numbers (Types) | Nouns, Raw Numbers (Types) | Source |
|---|---|---|---|---|
| Archi (aqc) | HF | 362 | 2419 | dictionary |
| Basque (eus) | HF | 1017 | 4707 | UD corpus |
| Georgian (geo) | HF | 25,467 | 92,691 | Georgian National Corpus |
| Hindi (hin) | HF | 601 | 6463 | UD corpus |
| Hungarian (hun) | HF | 2765 | 18,799 | Hungarian National Corpus |
| Japanese (jpn) | HF | 361 | 8952 | NINJAL |
| Korean (kor) | HF | 5308 | 33,172 | Seojeong Corpus, grammatically tagged |
| Latin, Classical (lat) | HF | 700 | 4777 | Aronoff (1994); Minozzi (2009) |
| Persian (per) | HF | 184 | 5163 | UD corpus |
| Tsez (ddo) | HF | 506 | 3508 | dictionary (Khalilov 1999) |
| Halkomelem (hur) | HI | 916 | 967 | dictionary (Galloway 2009) |
| Irish, Modern (gle) | HI | 890 | 1850 | Corpas na Gaeilge Comhaimseartha |
| Malagasy (mlg) | HI | 3643 | 5436 | dictionary (Diksionera 1973) |
| Māori (mao) | HI | 1656 | 2920 | dictionary (Williams 1957) |
| Zapotec (zap) | HI | 439 | 542 | dictionary (Long and Cruz 1999) |
| Zinacantec Tzotzil (tzo) | HI | 850 | 1629 | dictionary (Haviland 1994) |

Some of the languages in our sample are the same as reported by Polinsky (2012); however, some quantitative data are different (for example, for Hungarian, Japanese, and Korean). The differences between Polinsky's data and data reported here have to do with differences between WordNet (which Polinsky used for a number of languages in her sample) and corpora that we use here. The differences are not major, and the trends we present below largely replicate Polinsky's results. Nevertheless, we believe that the numbers presented here are more accurate because they reflect uniform lemmatization and rely on corpus/dictionary data. In fact, some of the problems that Polinsky (2012) faced and that were raised by Seifart et al. (2014) and Strunk et al. (2015) with respect to Polinsky's methodology were eliminated, here, because of a more uniform approach to data.

### 4.2. Testing the Predictions

Recall our assumption that the number of nouns should be relatively stable across different languages. Indeed, our data support this assumption. The percentage of nouns across the languages that we have discussed is relatively stable. In the corpora we consulted, the percentage of nouns is somewhere between 30 and 40 percent of the entries.[9] In the dictionaries, nouns constitute somewhere between 50 and 60 percent of the entries. That is a pleasing result, one that allows us to compare languages on a relatively stable footing. What really varies is the ratio of verbs to nouns. Languages with the lower ratio of nouns to simplex verbs are verb-rich, and the ones where nouns overpower simplex verbs, which leads to a higher verb-to-noun ratio, are verb-poor.

In order to test whether the verb-to-noun ratio is correlated with language headedness, we fit a logistic regression model, with language type as the only predictor. Treatment coding was used, with HF as the base level. The model's output is presented in Table 4 and the results are shown in Figure 1.

**Table 4.** Model coefficients.

| Coefficient | Estimate | Std. Error | z Value | $p$ |
|---|---|---|---|---|
| (Intercept) | −1.58 | 0.006 | −277.43 | $<2 \times 10^{-16}$ |
| HI | 1.04 | 0.013 | 78.64 | $<2 \times 10^{-16}$ |

We can convert the coefficients to average verb-to-noun ratios as estimated by the model by exponentiating them.[10] The model fit suggests that the verb-to-noun ratio is approximately 0.2 (1:5) in HF languages and approximately 0.59 (1:1.7) in HI languages. Given the magnitude of the *p*-values (very close to 0), these conclusions hold even if we wish to remain conservative and apply Bonferroni correction for multiple comparisons.

These results confirm the prediction that the number of lexical (simplex) verbs in head-final languages is significantly lower than in head-initial languages. To turn this around, upon encountering a lexical item in a head-final language, the likelihood of it being a noun rather than a verb is quite high; no such likelihood is available for head-initial languages. The results are not statistically different for rigidly head-final languages (Japanese and Korean) and those that are more flexible (Archi, Basque, Georgian, Hindi, Hungarian, Persian, Tsez, and particularly Classical Latin); all these languages cluster together as instantiating the head-final type.[11]

---

9　Persian seems to be an outlier in the UD corpora; in the Persian counts, nouns are at 62%. That may have something to do with the relatively small corpus size or with the particulars of morphological analysis for Persian. We do not have a clear explanation of this fact.

10　The exponentiated coefficient for the intercept will correspond to the average verb-to-noun ratio for the base level (in this case, HF); the exponentiated coefficients for the other levels of the predictor will correspond to the proportional change in the verb-to-noun ratio change relative to the base level.

11　These results pose an interesting challenge to existing analyses of Hungarian and possibly, other languages. Although Hungarian is described as head-final in typological literature (e.g., Dryer 2013a), Kiss (2002) considers its head-final order

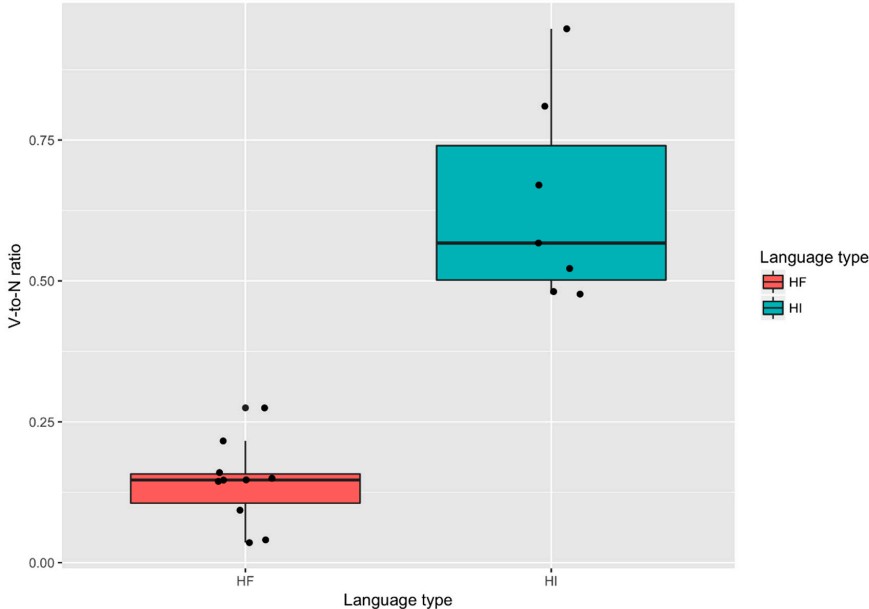

**Figure 1.** Verb-to-noun ratios, head-final (HF) and head-initial (HI) languages.

*4.3. The Pattern of Results: Verb-Rich and Verb-Poor Languages*

Our sample of languages with the lowest verb-to-noun ratios includes Archi, Basque, Georgian, Hindi, Hungarian, Japanese, Korean, Latin, Persian, and Tsez. All of them have ratios of around 0.2, and we characterize them as "verb-poor". With the exception of Latin, these languages are reliably head-final types, with postpositions and strictly head-final order in non-finite clauses. Japanese, Korean, Archi, and Tsez have prenominal relative clauses; Basque has both pre- and postnominal relatives, and Hungarian, Georgian, and Persian have postnominal relative clauses. Hindi has a rich variety of options, prenominal, postnominal, and correlatives (Mahajan 2000). The word order in the matrix (root) clause of these languages is actually more varied. While Korean and Japanese are rigidly head-final languages, the other verb-poor languages allow variation in the root clause, in particular, with the alternation between SOV and SVO. However, the uniformity of this language group with respect to the verb-to-noun ratio further suggests that the order of constituents in a matrix clause is just one of the many criteria of headedness, and also one whose reliability is undermined by information-structural and prosodic factors (some of which are not even well-known).

At the other end of the scale, we find languages with extremely high verb-to-noun ratios, on the average, 0.59 verb-rich languages. This set includes Halkomelem, Irish, Malagasy, Maori, Tzotzil, and Zapotec. All are consistently head-initial and have verb-initial orders as either the main order or one of the dominant orders.

So far, we have concentrated on languages that clearly conform to the head-final or head-initial type, and the order of constituents in the matrix clause of these languages attests to that, i.e., they are either verb-final or verb-initial. What about SVO languages? Researchers often take the SVO order in a matrix clause as a sign that the respective language is a head-initial type. The logic probably goes like this, it is not verb-final, hence not head-final, so it must be head-initial. If one follows this logic the prediction that can be made is that SVO languages should pattern with Halkomelem or Irish. In the next section, we explore this prediction.

---

derived from a head-initial base. Positing a particular underlying word order is often associated with specific theoretical assumptions, and those may not be shared by all other researchers. It is however possible that our generalizations reflect the output of particular syntactic derivations, and that output can be arrived at by different routes.

## 5. What about SVO Languages?

For a random sample of SVO languages, we selected seventeen languages with a wide geographical distribution, different morphological typology, and different genetic relations, although several families are represented by more than one language. These languages are shown in Table 5.

**Table 5.** SVO languages: Verb-to-noun ratios.

| Language | Verbs, Raw Numbers | Nouns, Raw Numbers | Source |
|---|---|---|---|
| Arabic (ara) | 1551 | 4662 | UD |
| Bobangi (bni) | 3324 | 3973 | dictionary |
| Bukusu (bxk) | 1653 | 2879 | dictionary |
| Chinese, Mandarin (chi) | 3430 | 18,764 | Tao and Xiao (2007) |
| Dutch (nld) | 2436 | 10,152 | UD |
| English (eng) | 2102 | 6342 | UD |
| French (fra) | 302 | 459 | UD |
| German (deu) | 2669 | 18,559 | UD |
| Greek, Modern (ell) | 956 | 2533 | UD |
| Hebrew, Modern (heb) | 1861 | 4202 | UD |
| Indonesian (ind) | 2742 | 5754 | UD |
| Polish (pol) | 3519 | 6238 | UD |
| Russian (rus) | 7832 | 16,761 | UD |
| Spanish (spa) | 5034 | 7652 | Corpus del español actual |
| Swahili (swa) | 3853 | 6150 | dictionary |
| Thai (tha) | 20,415 | 26,150 | dictionary |
| Vietnamese (vie) | 20,766 | 21,879 | Pham et al. (2008) |

Figure 2 below shows the distribution of all the languages in our sample with respect to their verb-to-noun ratios: the definitely head-final and definitely head-initial languages (see Table 3 and Figure 1) and SVO languages from Table 5.

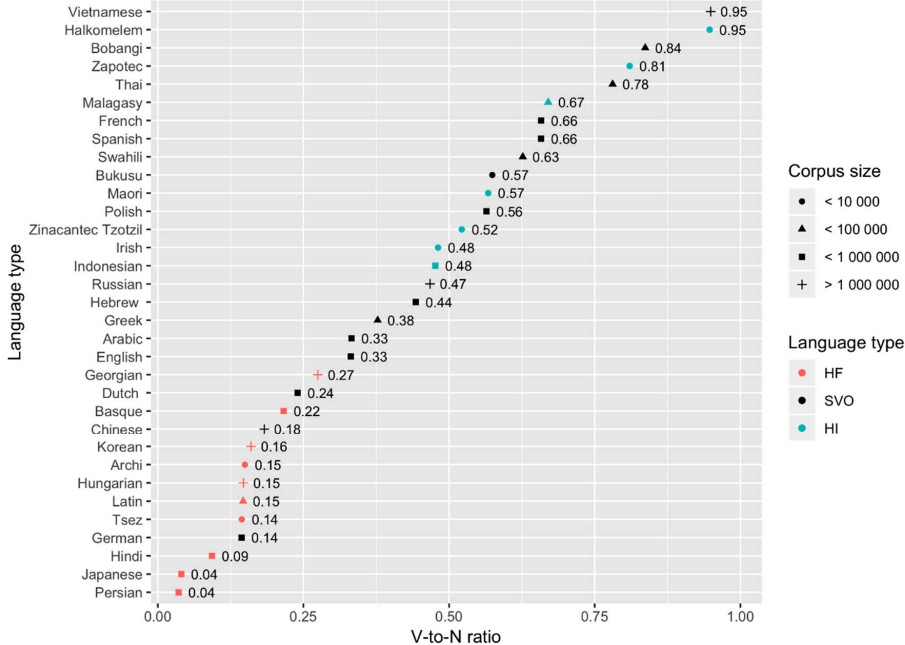

**Figure 2.** Verb-to-noun ratios: head-final languages, head-initial languages, and SVO languages (total 33 languages). Corpus size indicated with the dot shape.

The next visual representation (Figure 3) confirms what can be gleaned from the raw data, that is, verb-to-noun correspondences across SVO languages in our sample suggest that these languages do not form a uniform class.

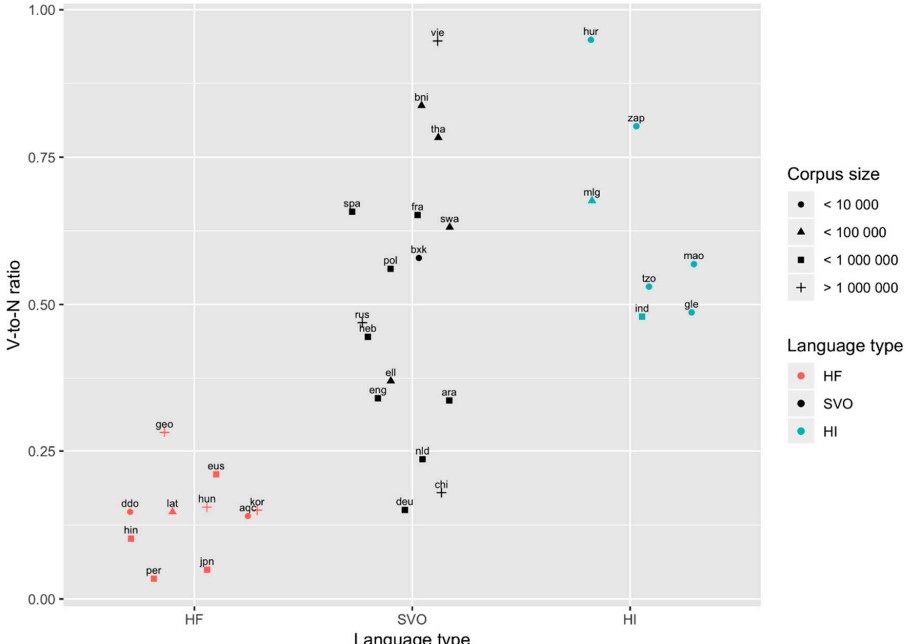

**Figure 3.** Verb-to-noun ratio by language type (HF, head-final; HI, head-initial; SVO, subject-verb-object). Three-letter language codes are given according to the ISO codes. Corpus size is indicated with the dot shape.

Three of the SVO languages are clearly close to the head-final type, with the comparable verb-to-noun ratio (German, Dutch, and Mandarin Chinese).

Next, a number of languages in the SVO sample pattern with the head-initial type; in fact, for some of them, the verb-to-noun ratio appears even higher than in the definitively head-initial languages, at 0.77; this group includes the three Bantu languages (Bobangi, Bukusu, Swahili), the two Romance (French and Spanish), Vietnamese, Thai, and Polish.

Finally, several languages in the sample are a bit closer to the head-initial type but stand sufficiently apart from them, which suggests that they can be counted as a separate group. Their verb-to-noun ratio is between 0.3 and 0.5; if we adopt 0.3 as the highest ratio for the clear head-final type, and 0.5 as the cut-off point for the head-initial type, the third group in SVO can be justified. These languages include Arabic, English, Modern Hebrew, Indonesian, Modern Greek, and Russian. For the purposes of the statistical analysis below, we have represented them as a separate group (M[iddle]), but this representation is not categorical, it is just a matter of convention. If one wished to have just two groups, the cut-off point could be shifted or eliminated. If more languages are tested, we could have firmer grounds for deciding on the number of headedness groups.[12]

Turning to the statistical analysis, as in the previous model (Section 4.2), we can convert the coefficients to average verb-to-noun ratios as estimated by the model by exponentiating them.[13] All pairwise comparisons are statistically significant,[14] indicating that languages with different headedness type do differ in their verb-to-noun ratios. Given the magnitude of the p-values,

---

[12] We will return to the headedness properties of the Middle group shortly.

[13] The exponentiated coefficient for the intercept will correspond to the average verb-to-noun ratio for the base level (in this case, HF); the exponentiated coefficients for the other levels of the predictor will correspond to the proportional change in the verb-to-noun ratio change relative to the base level.

[14] The z-value for the comparison between HF and M is not reported in Table 6; it can easily be obtained by refitting the model with M as baseline level. The following R command will do it: glm(cbind(v_count, n_count) ~ relevel(langtype, "M"), data = data, family = "binomial").

these conclusions hold even if we wish to remain conservative and apply Bonferroni correction for multiple comparisons (correcting for four comparisons across the two models).

Let us now examine the subgroups in the SVO group more closely. As we already mentioned, German, Dutch, and Mandarin Chinese cluster with the verb-poor group. And indeed, these languages have a number of strong head-final properties; for example, they allow the alternation between SVO and SOV (Dryer 2013a), have head-final orders in some embedded clauses, allow object shift and scrambling in the preverbal domain, which are typical of head-final languages. These phenomena are unattested or rare in those SVO languages that have head-initial properties (Dryer 1991; Vikner 1994; Biberauer and Roberts 2005, 2009; Sheehan et al. 2017, a.o.). Next, Mandarin Chinese, a source of never-ending sorrow for advocates of well-behaved SVO languages, exhibits prenominal relatives, which are very unusual in SVO languages.[15] In terms of its verb-to-noun ratio, Mandarin Chinese is very close to the one observed for head-final languages in our sample.

Several languages in this sample have ratios that group them together with the clearly head-initial languages discussed in the preceding section. This result is consistent with a number of head-initial properties observed in their structure. For instance, in Bantu, unlike in English, adjectival modifiers, possessive pronouns, and numerals follow rather than precede the head noun. Similarly, adverbs follow rather than precede the verb. The structure of teen numerals (12 ot 19) is "ten-with-unit" rather than "unit-ten". Tense, negation, and agreement are expressed by prefixation, not suffixation. All in all, the order of constituents in Bantu languages is a mirror image of what is found in the SVO/SOV languages (Dryer 2013a). The following are some examples illustrating the differences these properties in Swahili (irrelevant grammatical details are omitted):

(5)   *Swahili*

a.      Order of noun and adjective

    *safari*    *njema*
    trip      good

   "good trip; bon voyage"

b.      Order of noun and possessive modifier

    *jina*    *langu*
    name    my

   "my name"

c.      Order of noun and numeral

    *matunda*   *manane*
    fruit       six

   "six pieces of fruit"

d.      Order of verb and adverb

    *njoo*     *haraka*
    come.IMP quickly

   "come quickly"

e.      Structure of teen numerals

    *kumi*    *na*     *saba*
    ten      with    seven

   "seventeen"

---

[15]   As a result, researchers are often at a loss as to how to fit Mandarin Chinese in the word order typology (Dryer 1991, pp. 447, 476 for different, sometimes even conflicting approaches).

f.   Inflected verb

   *hawa-ta-fanya*
   3pl.neg-fut-make

   "they won't make".

The fact that the three Bantu languages, the two Romance languages, Thai, and Vietnamese pattern with the clearly head-initial languages, in the sample used here, supports the conception that the structure of the lexicon (as represented through the proportion of nouns to simplex verbs) reflects general headedness properties of a language rather than just the basic word order in a root clause.

The relatively high verb-to-noun ratio in Polish, especially given the contrast between Polish and Russian (the ratios are 0.56 and 0.47, respectively), could be a reflection of the way corpora or dictionaries are organized (we flagged this issue in our discussion of methodological limitations and difficulties, in Section 3). Here we agree with Polinsky (2012, pp. 354–55) that differences in verb counts could be due to:

> the outstanding issues that Slavic lexicographers need to deal with: verb aspect; reflexive verbs; verb prefixation (single, double, triple) . . . . For instance, the number of verbs could go up or down depending on how the lexicographer approaches Slavic aspectual pairs: does one count verbs in the perfective and imperfective as separate lemmas or as members of the same lemma? Counting all verbs twice obviously inflates the size of the verbal lexicon . . . . These . . . factors alone are more than sufficient to force an even greater discrepancy than the one we observe.

Finally, English, Arabic, Hebrew, Indonesian, Greek, and Russian are considered genuinely in the middle, with their verb-to-noun ratios positioned between well-behaved head-final languages and their head-initial polar opposites. Earlier in this section we made an apparently opportunistic move by separating them from the clearly head-initial group on the basis of the verb-to-noun ratios; this separation as the Middle subgroup was supported by the statistical analysis (see Table 6). However, as we look at the headedness of these languages, comparing them, for example, with Bantu, we observe a number of patterns that are a mirror image of Bantu such as ordinary and possessive modifiers typically precede the head noun (albeit not in Semitics), the numeral precedes the quantified expression in a numerical phrase, the order within numerals is "unit-ten", and usually there are no clause-initial question particles. Thus, the headedness profile of the middle subgroup is somewhat different from the head-initial subgroup in the SVO sample, and division between these two subgroups is less arbitrary than it may seem.

**Table 6.** Model coefficients for the model based on the full dataset (33 languages).

| Coefficient | Estimate | Std. Error | z Value | *p* |
|---|---|---|---|---|
| (Intercept) | −1.61 | 0.005 | −313.57 | $<2 \times 10^{-16}$ |
| HI | 1.31 | 0.007 | 181.33 | $<2 \times 10^{-16}$ |
| M | 0.79 | 0.011 | 74.54 | $<2 \times 10^{-16}$ |

As is often the case in structural typology, genetic relations are easily overridden by structural differences. For example, English is genuinely SVO, but German and Dutch look emphatically head-final/OV. Likewise, classical Latin is head-final/OV, patterning with the verb-poor group, but Spanish and French have moved decidedly toward the VO type.

The heterogeneity of the SVO group is consistent with the conception that languages with the basic SVO order do not represent a uniform type. Likewise, the data shown above indicate that SVO does not equal head-initiality. Rather, languages with the SVO order in the main clause subsume OV and VO languages (Dryer 2013e). Many researchers agree that OV and VO are simply representations

of head-final and head-initial structures, respectively (Lehmann 1973, 1978; Vennemann (1974, 1976), for the initial proposal, and Dryer (2013e) for discussion and examples). Each subtype has significant structural corollaries, and the verb-to-noun ratios observed, here, are consistent with the other characteristic features of the relevant languages.

It is tempting to draw more fine-grained distinctions within the SVO group, but at this point, it is more important to pause and try to make sense of the data on a macro level. The clear distinction that emerges from these data is between VO languages, whose verb-to-noun ratio is quite high, meaning that they are verb-rich, and OV languages, which have a low(er) verb-to-noun ratio and which are verb-poor. Even if we assume that there is variation in our sample (something we will discuss more in the Section 6), we still find a significant clustering of verb-poor languages in the head-final type and of verb-rich languages in the head-initial type.

To return to our original question in (1), the query was whether or not the ratio of nouns to lexical verbs (as opposed to LVCs) depends on headedness. On the basis of the data presented here, the answer to this question is positive. Our results show a clear correlation between headedness and the proportion of lexical (simplex) verbs in the lexicon. Head-initial/VO languages (Irish, Malagasy, Maori, Tzotzil, Bantu, and Thai) have a particularly high proportion of such verbs. In contrast, languages of the rigidly head-final type are verb-poor and compensate for that by a heavy use of LVCs.

What can account for the observed relationship between headedness and size of the verb lexica? In the next section, we offer some considerations that help explain this pattern of headedness.

## 6. In Search of an Explanation

### 6.1. General Remarks

We have so far established a correlation between headedness and the number of lexical verbs in a given language, as revealed in the verb-to-noun ratio across a sample of languages. The proportion of nouns in a lexicon seems to be relatively stable cross-linguistically, and variation is observed in the verb-to-noun ratio for a particular language. This ratio is quite low in head-initial languages, which we characterized as verb-rich, and high by comparison in head-final languages. Languages with the SVO basic order fall into different subclasses; the OV subtype patterns with head-final languages, some VO languages are close to the head-initial type, and some are more or less in the middle (English, Russian).

One way to make sense of these results is to tie them to the possible or preferred derivational morphology used by a given language. In that case, the correlation is not between headedness and the size of the nominal/verbal lexicon but, instead, between headedness and strategies for deriving new verbs. Looking back at the examples used in this paper, we have seen that English happily zero-derives verbs, Russian adopts new verbal roots with or without a derivational suffix, and head-final languages prefer to use light verbs. As a result, each subtype ends up with more or fewer simple verbs. There is a reasonable correlation to be drawn here, but it ultimately leads to a restatement of the facts rather than an explanation for them. Furthermore, if we investigate a particular morphological subtype, for example, languages with impoverished morphology, which extensively use zero derivation, the results are mixed. Both English and Mandarin Chinese rely on zero derivation (see Xue 2001 for Mandarin Chinese), but their headedness properties are quite different.

Thus, morphological differences do not lead us to a satisfying explanation. Before we explore alternatives, we ask two more general questions concerning the possible baseline and the causal relationship between headedness and the size of the verbal lexicon.

### 6.2. Establishing the Baseline

The first question that we need to resolve is as follows: Which part of the asymmetry in headedness needs to be explained, the rich inventory of lexical verbs in the head-initial type, or the dearth of such verbs in head-final languages? and secondly, What is the causality in the relationship between

headedness and the number of verbs? and Is headedness a response to the structure of the lexicon, or is the composition of the verbal lexicon an adaptation to headedness?

To address the first question, it is useful to consider the distribution of head-final and head-initial orders in language structure. To recapitulate the structural generalizations brought up earlier, head-final orders allow more freedom of distribution than head-initial ones. Recall Greenberg's tetrachoric universals, which all impose constraints on the head-initial design, as well as the FOFC, which limits the distribution of head-initial structures as well. That suggests that head-final structures are somehow more optimal, or to turn the tables around, head-initial structures are more restricted. These structural considerations seem to be supported by the cross-linguistic data concerning the distribution of different orders. Cross-linguistically, head-final orders are more common. Dryer (2013a) lists 569 verb-final languages; in contrast, only 120 languages are verb-initial and 499 languages, in his sample, are verb-medial in which he identifies only 29 as SOV/SVO. However, even if half of the "SVO" type in his sample have OV properties, that shows significant skewing in favor of the head-final type.[16] Based on this logic, it makes sense to consider head-final/OV structures as the starting point.

The next question has to do with causality in the correlation between the head-final design and paucity of lexical verbs. Do head-final/OV languages have few lexical verbs because they are head-final, or is the dearth of lexical verbs an adaptation to the head-final type? To address this question, we turn to structural heads rather than verbs. If it were part of the head-final design that the inventory of heads be smaller than in non-head-final structures, we would also expect a smaller set of structural heads in other categories, in particular, in adpositions.

Unlike verbs, adpositions form a closed class of items; they typically express temporal or spatial relations and can also index the thematic role of their complement (which is typically expressed by a noun phrase). Adpositions comprise prepositions and postpositions. Prepositions appear before their complement noun phrase, and postpositions follow their complement. As with other Greenbergian tetrachoric universals, the inviolable condition states that head-initial languages are not expected to have postpositions. Head-final languages can have postpositions (typical of rigid head-final languages) or prepositions. Table 7 below reflects this condition.

**Table 7.** Adpositions across language types.

|  | Head-Final Type | Head-Initial Type |
| --- | --- | --- |
| **Preposition** | possible | possible |
| **Postposition** | possible | impossible |

As is often the case, languages with the SVO basic order do not all conform to this condition. Those SVO languages that have strong head-initial properties, lack postpositions, but SVO languages with OV properties can have them, although sometimes they are more relics of the past. For example, the English *ago*, *later*, and *aside* are considered postpositions, and *notwithstanding* is an ambiposition (Hagège 2010) which can appear before and after its complement. In Russian, the semantic equivalent of *ago*, the word *nazad*, is a strict postposition, and there are a fair number of ambipositions (Philippova 2018).

Back to the causal relation between headedness and the size of a lexical category. If the content of a category determined headedness, we would expect postpositional classes to be smaller than prepositional classes. If headedness were independent of the class content, we would not expect such a difference. To test this hypothesis, we examined the number of prepositions vsersus the number of postpositions across a set of languages.

---

[16] We would like to underscore that this cross-linguistic distribution should be taken seriously primarily because of the structural principles just mentioned. Without those principles, it could have been less dramatic and could be due to historical accidents or extralinguistic factors.

Before we present our results, some general remarks are in order. Although adpositions form a closed class, the boundaries of that class are not always clear. Several factors seem to contribute to the uncertainty. First, the division between cases and adpositions is not always well-defined (Blake 1994; Kracht 2002; Asbury 2008; Trommer 2008). For example, there is an ongoing debate among researchers working on Japanese whether exponents *-ni* [DATIVE] or *-e* "to" are case markers, postpositions, or both (Miyagawa 1989; Watanabe 2009). In another example, consider Baker and Kramer (2014) on apparent prepositions in Amharic, which they convincingly analyze as case morphemes. A second set of controversial instances often include adverbials that can be reported to occasionally pattern as adpositions (Marácz 1989; Kiss (2002), for the relevant distinctions in Hungarian, and Van Riemsdijk (1997) for a more general discussion). Next, some adpositions are complex. For instance, Amharic grammars list the four postpositions below, which are all formed from the apparent preposition *wädä* "towards" and the demonstrative pronouns (*yih, ya, innäzzih, innäzziya*). In counting lemmas, it would not make sense to distinguish four different items.

(6)　*Amharic*

a.　wädiya;
b.　wädäzziya;
c.　wädäzzih;
d.　wädih.

Continuing on the theme of composite items, some apparent adpositions are in fact relational nouns, not dissimilar from the English *front* in *in front of*. Again, their attribution to the adpositional class is questionable.

With these confounds in mind, we compiled a random language sample (30 languages, Appendix A), using several different means. We relied on grammars, data collection, corpora (UD tagged, as in the data on verb-to-noun ratio), and additional verification based on research on the respective languages.[17] We included in our study only those items that were identified as adpositions in individual grammars, trying to exclude all controversial instances.

Of the languages included in this sample, twelve had only prepositions (Pr),[18] thirteen had only postpositions (Po), and five had both. We ran a Pearson correlation on the number of prepositions (Pr) versus the number of postpositions (Po) in each language within our sample. We found no significant correlation (rho = 0.47 and $p = 0.09$) across the sample with respect to the number of Pr and the number of Po. We also ran a mixed log-linear model (Poisson regression) predicting the number of distinct adposition types from their position in the noun phrase. We found no significant difference in the overall number of adpositions ($\beta = -0.31$, $z = -0.35$, and $p = 0.73$), even when languages that lack a certain kind of adposition were completely excluded from the analysis ($\beta = 0.1$, $z = 0.34$, and $p = 0.74$). See Appendix B for details of the statistical analysis.

In languages that have both types of adpositions, asymmetries can go either way. In particular, German has more prepositions than postpositions, but in Armenian the relationship is the other way around. Over the history of Armenian, the balance has shifted from preposition-heavy in Classical Armenian to postposition-heavy in Modern Armenian (possibly under the influence of Turkish).[19]

Even though our sampling methods can be taken to fault, the result appears quite straightforward, that is, there is no difference between the head-final and head-initial (or to be more precise, not-head-final) type in the size of the adpositions class. According to this result, we conclude that headedness does not affect the size of the class of adpositions.

---

[17]　In the UD corpora, the reliability of tagging may vary—it is as good as the underlying analyzer is. For example, one of the tagged corpora for Japanese, GSD, lists 63 lemmas for postpositions (tag ADP), and another, corpus BCCWJ, lists one!
[18]　We counted English and Russian, which have very few postpositions, in the Pr-language class.
[19]　We would like to thank Bert Vaux for this example of historical change and helpful discussion.

These results suggest that the dearth of lexical verbs in head-final languages is a consequence, not a cause, of headedness which is an adaptation made under a particular structural design. Why? We offer some considerations in the next subsection.

*6.3. Where Have All the Verbs Gone?*

Informativity or informational contribution of a linguistic element is defined as the degree of uncertainty associated with that element, namely, whether what follows that element can be expected and to what extent (this expectation includes the possibility that the element is stand-alone). Informativity is not limited to word-order principles. In the following simple example, consider the use of /ʒ/ and /s/ in the word-initial position in English where /ʒ/ is extremely informative; only a handful of English words start with it, and those are all loanwords (*genre, jabot, jus*).[20] In contrast, the initial /s/ is uninformative; a great number of English words start with it, therefore, its appearance does not reduce the uncertainty of what follows. In a different example, the high back rounded vowel /ʉ/ is extremely uncommon in the word-initial position in American English (*Uber, ooze, oodles, Uzbek*), which makes it highly informative, reducing the uncertainty of what follows it. Contrast this sound with the initial /i/ in American English; this word onset is very common, which leads to uncertainty as to what follows it, making it uninformative.

If we now return to the order of elements in a clause, the first word of a clause is informationally most uncertain, since it can allow many possible continuations. As the clause progresses, the closer the parser is to the end of the clause, the less uncertainty there is; thus, there is less pressure to have a variety of words at the end of the clause. In other words, the difficulty in understanding and predicting the next word in a clause should be decreased the farther one goes, and that accounts for the relatively small number of lexical verbs in OV languages. Turning this around, the verb, which has to come at the end of an SXV clause, should carry less information because it comes so late, and it makes sense to frontload the more contentful constituent which includes the non-verbal part of an LVC. Although the material preceding the light verb is typically verb-adjacent, its position before that verb appears to make a difference.

The small number of lexical verbs in head-final languages is not the only "adaptation" to headedness found in such languages. Head-final languages are also characterized by the intransitive bias; they have a much higher proportion of intransitive verbs, which reduces the distance that has to be traversed in the clause before the parser reaches the predicate (Ueno and Polinsky 2009). The intransitive bias serves to reduce the argument-assigning domain by having fewer verbal complements, and in OV languages it allows one to reduce the number of preverbal arguments.

In light of this logic, the heavy use of LVCs in head-final languages no longer appears to be accidental. Instead, it is yet another way of increasing informativity of a clause before the verbal part of the predicate is reached. We would like to underscore that this is not a bidirectional correlation; while head-final languages tend to have a high incidence of LVCs, LVCs can be motivated by other factors, and their occurrence is not limited to head-final languages (Amberer et al. 2001; Brugman 2001; Folli and Harley 2013, a.o.).

These observations allow us to make further predictions. If a language shifts away from the head-final type, we expect it to develop more lexical verbs. Conversely, if a language becomes a more strictly head-final type, its lexical verb class can start shrinking. Support for the former prediction can actually be found in our language sample; Latin has a relatively low verb-to-noun ratio, indicative of its OV/head-final properties, whereas the descendant Romance languages (Spanish, Romanian) have a higher verb-to-noun ratio, with a larger class of lexical verbs. With respect to the latter prediction, we do not have data to test in in our sample. Hopefully such languages will be found in future studies.

---

[20]   In fact, many speakers of American English strengthen this sound to [dʒ] in the onset position.

Several questions arise at this point. First, why do verbal structures have to undergo an "adaptation" while adpositional structures do not? Recall that we found no difference between head-initial/VO languages and head-final/OV languages in the size of their adpositional class. We contend that the difference has to do with the complexity of the corresponding phrase. Adpositions take only one complement, so the uncertainty in an adpositonal phrase is below the threshold of difficulty. Verbs can take more than one complement and project specifiers, and therefore the complexity of a verb phrase is greater than that of an adpositional phrase and that creates more need for reducing informational uncertainty.

The second question we would like to raise takes us back to the distribution of headedness types. If the unequal distribution of verbs in head-final/OV languages is an adaptation or a "correction" that these languages make because of their design, why is this type so common? And conversely, why are strictly head-initial languages, where the verb is the starting point, so uncommon?

After all, nouns and verbs are not equal with respect to their informativity. Since verbs take arguments, their relational structure is richer than that of most (non-relational) nouns (Gentner 1981; Langacker 1987, 2002, chp. 3; Baker 2003). Although different researchers implement the basic idea differently, they all converge on the conception that verbs express relations, thus, bringing in associated arguments or scenes, whereas nouns denote what Langacker calls "regions", isolated entities which typically do not bring in associated arguments, of course, relational nouns are an exception to that. Thus, on the whole, verbs are semantically more informative than nouns. Next, an overt finite verb is necessary to form a clause; pro-verbs are extremely rare, whereas pronouns, including silent ones, are extremely common across languages (Seifart et al. 2018 for similar observations). Accordingly, verbs are both informative and structurally necessary because of their connection to the inflectional head of a clause. The semantic and grammatical complexity of verbs is supported by the increased processing costs associated with verbs as compared with nouns (Vigliocco et al. 2011 and references therein). Why not, then, use verb first? By this logic, we should all be speaking verb-initial languages. The answer to this question has to do with the general principles of anchoring, the segmentation of scenes into figure and ground (Talmy 1975, 2000), and the related division of judgments into thetic and categorical (Kuroda 1972; Sasse 1987). In other words, headedness plays a role in language design, but it is not the sole factor, and it should be evaluated more broadly in the context of semantic, information-structural, and parsing properties.

## 7. Conclusions

The initial question we pursued, in this paper, concerned whether the ratio of nouns to lexical verbs differs meaningfully across headedness types. We collected quantitative data on the verb-to-noun ratio across a sample of languages, chosen more or less opportunistically as a "convenience" sample (focusing on languages for which it was possible to obtain reliable numerical data on the size of the nominal and verbal class).

The following results are surprising: There is a robust correlation between headedness and the proportion of lexical (simplex) verbs in the lexicon. Head-final/OV-type languages have a relatively small percentage of simplex verbs, whereas head-initial languages have a considerably larger percentage of such verbs. The OV/VO difference with respect to verb-to-noun ratios also reveals itself in SVO languages; some languages, for example, Mandarin Chinese, German, and Dutch among them, show the verb-to-noun ratio characteristic of the head-final/OV type, whereas others, such as Romance or Bantu, are head-initial/VO-like in their verb-to-noun ratio which is consistent with a number of their structural properties.

The proportion of verbs to nouns, thus, emerges as a new linguistic characteristic that is correlated with headedness. Further empirical verification is needed, and assuming that further studies confirm this new generalization, the connection between lexicon and syntax is tighter than it is usually assumed.

The relationship between headedness and the number of lexical verbs is well supported by quantitative data. However, the reasons for the asymmetry in the verbal lexicon between head-final

and head-initial languages are less clear. In this paper, we proposed an initial and tentative explanation for the quantitative facts we have reported. In head-final languages, the verb, which encodes the main predicate, appears at the end of its clause, and since it comes so late, it is expected to be more general and carry less specific information. The common use of LVCs in head-final languages no longer appears to be accidental. Instead, it is yet another way of increasing informativity of a clause before the verbal part of the predicate is reached. We would like to underscore that our tentative explanation is rooted in fundamental conceptual differences between nouns and verbs, which has been explored by a number of researchers of different theoretical persuasions (most notably, Gentner 1981; Baker 2003). This explanation should be explored more in depth if one wants to maintain that there exists a strong connection between the syntax and the lexicon.

**Author Contributions:** M.P. is responsible for the general conception of the paper, the collection of initial data, general discussion, and the write up of the paper; L.M. is responsible for the data on adpositions and general discussion. All authors have read and agreed to the published version of the manuscript.

**Funding:** This material is based upon work supported by the National Science Foundation under grant BCS-1619857 to the first author.

**Acknowledgments:** For comments on this paper and help with data collection, we are grateful to Eneko Agirre, Archna Bhatia, Francis Bond, Leston Buell, Peter Graff, Verena Hinrichs, Larry Hyman, Peter Jenks, Mikhail Kopotev, Karine Megerdoomian, Keith Plaster, Victor Raskin, Kevin Ryan, Barbara Stiebels, Dan Tufis, Bert Vaux, Shuly Wintner, Colin Zwanziger, and three anonymous reviewers. We also thank Anton Malko for help with statistical analysis.

**Conflicts of Interest:** The authors declare no conflict of interest.

## Appendix A Language Sample, Adposition Study

| Language | Pr and/or Po |
| --- | --- |
| Adyghe | Po |
| Amharic | Po |
| Armenian | Pr/Po |
| Basque | Po |
| Bulgarian | Pr |
| Dutch | Pr |
| English | Pr |
| Finnish | Pr/Po |
| French | Pr |
| Georgian | Po |
| German | Pr/Po |
| Gombe Fula | Pr |
| Hungarian | Po |
| Hunzib | Po |
| Ingush | Po |
| Irish | Pr |
| Japanese | Po |
| Korean | Po |
| Latin | Pr/Po |
| Lezgian | Po |
| Mandarin Chinese | Po |
| Marathi | Po |
| Modern Greek | Pr |
| Modern Irish | Pr |
| Musqueam | Pr |
| Ndyuka | Pr |
| Ossetic (Digor) | Pr/Po |
| Russian | Pr |
| Spanish | Pr |
| Turkish | Po |

## Appendix B Analysis of Adpositional Data

The model included random intercepts and slopes grouped by language.

**Table A1.** Analysis of adpositional data.

| Coefficient | Estimate | Std. Error | z Value | *p* |
|:---:|:---:|:---:|:---:|:---:|
| (Intercept) | 2.9485 | 0.1205 | 24.460 | $<2 \times 10^{-16}$ *** |
| Pr ~ Po | 0.1001 | 0.2977 | 0.336 | 0.737 (n.s.) |

Notes: *** indicates a highly significant effect.

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
