# Peer review of "Headedness and the Lexicon: The Case of Verb-to-Noun Ratios"

_languages, doi:10.3390/languages5010009_

Round 1

Reviewer 1 Report

This article explores the connection between headedness and the size of a lexical class, in particular the verb class. The authors use facts from 33 languages and show that (a) head final and head initial languages have comparable verb-to-noun ratios but (b) the former have a smaller number of simplex verbs than the latter and compensate for this by a heavy use of light verb constructions.

I think that the relationship between headedness and the number of lexical verbs in a language is well supported by the quantitative data offered in the article but, as the authors themselves acknowledge, the reasons for this asymmetry in the verbal lexicon are less clear. The explanation they offer is only tentative and should be explored more in depth if one wants to maintain that there exists a strong connection between the syntax and the lexicon. As argued at some point in the paper, headedness plays a role in language design, but other aspects such as information structure or parsing properties should also be taken into account when approaching word order and its potential influence on vocabulary. The first steps taken here in this respect are well motivated and seem rather plausible; I encourage the authors to advance in that direction.

The article is written clearly and correctly, and the guiding hypothesis is sharply defined. Argumentation is sound and the authors do not hesitate to address explicitly the implications (and complications) of every issue involved in the proposal. I think they show reasonable knowledge of previous scholarship with reference to the topic under discussion, but since the work does not fall within my field of expertise I cannot evaluate its contribution to already existing knowledge. Providing this contribution is significant, I do not have any objection to its publication in Languages as is.

Author Response

We are grateful to the reviewer for their helpful comments. Our replies are shown in boldface below. 

I think that the relationship between headedness and the number of lexical verbs in a language is well supported by the quantitative data offered in the article but, as the authors themselves acknowledge, the reasons for this asymmetry in the verbal lexicon are less clear. The explanation they offer is only tentative and should be explored more in depth if one wants to maintain that there exists a strong connection between the syntax and the lexicon.

We added some prose to the conclusion explaining the tentative nature of our explanation and the need to look into the connection between lexicon and syntax more closely.

As argued at some point in the paper, headedness plays a role in language design, but other aspects such as information structure or parsing properties should also be taken into account when approaching word order and its potential influence on vocabulary.

We agree completely, and we added a clarification to this point in the beginning of section 2.2

The first steps taken here in this respect are well motivated and seem rather plausible; I encourage the authors to advance in that direction.

The article is written clearly and correctly, and the guiding hypothesis is sharply defined. Argumentation is sound and the authors do not hesitate to address explicitly the implications (and complications) of every issue involved in the proposal. I think they show reasonable knowledge of previous scholarship with reference to the topic under discussion, but since the work does not fall within my field of expertise I cannot evaluate its contribution to already existing knowledge. Providing this contribution is significant, I do not have any objection to its publication in Languages as is.

Reviewer 2 Report

The paper is original and interesting, the results promising. But in major respects it is not rigorously argued enough to merit publication. The statement of the hypotheses, procedures of quantification and formulation of the findings all need to be made more precise.

The main problem is that the concept ‘number of verbs’ is not sufficiently clear. At several points it seems ambiguous between verb types (number of different verbs available in the lexicon) and verb tokens (number of verbs appearing in a sample). Do the hypotheses apply to types, tokens or both?

Methodologically, a concern is that the data mix dictionaries and corpora indiscriminately, as in Table 3. In principle, dictionaries are suitable only for verb types, corpora for tokens and arguably (subject to size of the corpus) also for types. Corpus size is included as a variable in Figures 2 and 3, but its significance is not discussed, nor is there any obvious pattern (perhaps this could be turned into an argument that corpus size does not matter, but this would be surprising and needs to be discussed). Nevertheless, if one ignores corpus size, the correlation between headedness and verb-to-noun ratio in Figure 3 is striking.

The title is not sufficiently informative, and needs to refer to the verb/noun ratio. Adding a sub-title such as ‘the case of verb to noun ratios’ would be one way to specify this.

The ‘anecdotal’ style of the introduction is unusual, but not unwelcome.

Should 3b be labeled ‘alternative hypothesis’? Also, ‘smaller number of simplex verbs’ is highly ambiguous and needs to be made more precise: for example, ‘a lower ratio of verb types to noun types’.

Section 3: ‘the ratio of nouns to verbs’ should read ‘the ratio of verbs to nouns’ for consistency with Table 2.

In Table 2 and the surrounding discussion it is not clear whether the figures are for verb types or verb tokens. It seems to be types, but this picture is confused by the column of tokens.

Section 4.1 seems to confirm that the hypotheses refer to verb types, but equivocates by saying ‘the general focus on types (rather than tokens) in this paper’.

‘comparable for our purposes’ should probably read ‘comparable with regard to verb types’

To test the hypotheses, it may be worth counting both types and tokens, based on dictionaries and corpora respectively. The results could bear on the explanation for the correlations.

Regarding pro-drop, there is a large literature on child language which is relevant. Work on English suggested that children’s early vocabulary showed a ‘noun bias’ (more nouns than verbs) but studies on Korean (Choi and Gopnik 1995) and Chinese (Tardif 1996, 1999) found no such bias in these languages. The typological difference is attributed in part to pro-drop.

In table 3, ‘raw numbers’ is again ambiguous between types and tokens.

In section 4.2, the percentage of nouns is calculated in terms of tokens.

p. 11 language type is used as a predictor, but only two values (Head-final and Head-initial) are recognized here. It has already been spelt out that there are rigid and non-rigid verb-final languages, so these should be treated separately. On p. 12 it is claimed that these sub-types do not differ with regard to the V-to-N ratio, but this cannot simply be asserted: it needs to be shown by running the model separately for the sub-types.

p. 12 ‘strictly head-final order in embedded clauses’ does not sound right for Hungarian.

p. 17 here it is noted that the number of verbs types for Slavic languages will depend on whether perfective and imperative forms are assigned to the same verb type or two (similar issues will arise with derivational morphology in other languages). But this acknowledgment is too little, too late: the procedure for counting verb types in dictionaries/corpora need to be clearly spelt out in the methodology section.

p. 22 ‘Latin has a relatively high verb-to-noun ratio’ would be the wrong result for the hypothesis. Presumably this should be ‘a relatively low verb-to-noun ratio’, while the modern Romance languages have a higher ratio.

Author Response

We are grateful for the comments and we have addressed them in the text and in the attached document, which shows what exactly was done in the revisions. Please see the PDF attachment for detailed replies. 

Reviewer 3 Report

The article is very well written and presents a new data set from 33 languages. The structure is clear; the argument unfolds in a logical way; the methodology is clearly explained. 

My suggestion is regarding the research question. First it is formulated in the abstract (line 5), but nothing is said about its origin: is it a new question? Any prior research on this topic? Later on, in part 1, lines 57 & 58, you mention that “we build and expand the initial generalizations proposed by Polinsky (2012), Seifart et al. (2014), and Trunk et al. (2015).” But did they explore the same question? If so, what do your findings show with respect to their generalizations? How did you contribute to the existing literature? To demonstrate your contribution, could you briefly explain in the abstract whether you propose a new line of inquiry or whether you provide an alternative answer to an existing question and how your work adds to the literature? 

Some minor observations:

Your results show the verb-to-noun ratio across languages, but in the lines 56, 205, 492, for example, it is "proportion/ratio nouns to verbs". Wouldn’t it be easier to interpret the results with the uniform verb-to-noun ratio?

Line 46: “to park” in Russian is “parkovat’,” the verb “parkirovat’” may not exist. 

Author Response

We are grateful for the comments below and we have tried our best to address them. Our responses are shown in boldface. 

The article is very well written and presents a new data set from 33 languages. The structure is clear; the argument unfolds in a logical way; the methodology is clearly explained. 

My suggestion is regarding the research question. First it is formulated in the abstract (line 5), but nothing is said about its origin: is it a new question? Any prior research on this topic? Later on, in part 1, lines 57 & 58, you mention that “we build and expand the initial generalizations proposed by Polinsky (2012), Seifart et al. (2014), and Trunk et al. (2015).” But did they explore the same question? If so, what do your findings show with respect to their generalizations? How did you contribute to the existing literature? To demonstrate your contribution, could you briefly explain in the abstract whether you propose a new line of inquiry or whether you provide an alternative answer to an existing question and how your work adds to the literature? 

We have added some prose in that regard and have modified the abstract.

Some minor observations:

Your results show the verb-to-noun ratio across languages, but in the lines 56, 205, 492, for example, it is "proportion/ratio nouns to verbs". Wouldn’t it be easier to interpret the results with the uniform verb-to-noun ratio?

Thank you for catching this, we have changed the wording to make it uniform.

Line 46: “to park” in Russian is “parkovat’,” the verb “parkirovat’” may not exist. 

We checked with three Russian speakers, one of them allowed this form. However since there is doubt we removed it and left only parkovat’.